# Treatment of the Neutropenia Associated with GSD1b and G6PC3 Deficiency with SGLT2 Inhibitors

**DOI:** 10.3390/diagnostics13101803

**Published:** 2023-05-19

**Authors:** Maria Veiga-da-Cunha, Saskia B. Wortmann, Sarah C. Grünert, Emile Van Schaftingen

**Affiliations:** 1Metabolic Research Group, de Duve Institute and UCLouvain, B-1200 Brussels, Belgium; 2University Children’s Hospital, Paracelsus Medical University, 5020 Salzburg, Austria; s.wortmann@salk.at; 3Amalia Children’s Hospital, Radboudumc, 6525 Nijmegen, The Netherlands; 4Department of General Pediatrics, Adolescent Medicine and Neonatology, Medical Center, Faculty of Medicine, University of Freiburg, 79106 Freiburg, Germany; sarah.gruenert@uniklinik-freiburg.de

**Keywords:** metabolite repair, glucose-6-phosphatase, neutrophil, infection, neutropenia, glucose-6-phosphate transporter, gluconeogenesis

## Abstract

Glycogen storage disease type Ib (GSD1b) is due to a defect in the glucose-6-phosphate transporter (G6PT) of the endoplasmic reticulum, which is encoded by the SLC37A4 gene. This transporter allows the glucose-6-phosphate that is made in the cytosol to cross the endoplasmic reticulum (ER) membrane and be hydrolyzed by glucose-6-phosphatase (G6PC1), a membrane enzyme whose catalytic site faces the lumen of the ER. Logically, G6PT deficiency causes the same metabolic symptoms (hepatorenal glycogenosis, lactic acidosis, hypoglycemia) as deficiency in G6PC1 (GSD1a). Unlike GSD1a, GSD1b is accompanied by low neutrophil counts and impaired neutrophil function, which is also observed, independently of any metabolic problem, in G6PC3 deficiency. Neutrophil dysfunction is, in both diseases, due to the accumulation of 1,5-anhydroglucitol-6-phosphate (1,5-AG6P), a potent inhibitor of hexokinases, which is slowly formed in the cells from 1,5-anhydroglucitol (1,5-AG), a glucose analog that is normally present in blood. Healthy neutrophils prevent the accumulation of 1,5-AG6P due to its hydrolysis by G6PC3 following transport into the ER by G6PT. An understanding of this mechanism has led to a treatment aimed at lowering the concentration of 1,5-AG in blood by treating patients with inhibitors of SGLT2, which inhibits renal glucose reabsorption. The enhanced urinary excretion of glucose inhibits the 1,5-AG transporter, SGLT5, causing a substantial decrease in the concentration of this polyol in blood, an increase in neutrophil counts and function and a remarkable improvement in neutropenia-associated clinical signs and symptoms.

## 1. Introduction

Glycogen storage disease type I (GSD1), or von Gierke’s disease, was the first form of glycogenosis to be described and also the first one for which the enzyme deficiency—glucose-6-phosphatase deficiency—was elucidated soon after the discovery of this enzyme [1]. The symptoms of this disease (reviewed in [2]), principally hepatomegaly (and nephromegaly), due to glycogen accumulation in the liver and the kidneys, together with hypoglycemia and lactic acidosis, are an obvious consequence of a lack of glycogen degradation and the conversion of gluconeogenic precursors to glucose in the two organs that are by far the most important ones for the production of glucose during fasting.

It took longer to realize that some rare cases of patients with a typical GSD1 metabolic symptomatology had normal glucose-6-phosphatase activity in liver biopsy specimens [3], leading to the idea that there were two forms of GSD1, GSD1a (OMIM #232200), which is due to a true glucose-6-phosphatase defect, and GSD1b (OMIM #232220), which is due to another, undefined defect. Progress in the biochemistry of the glucose-6-phosphatase system [4] next led to the understanding that glucose-6-phosphatase, well known to be a transmembrane protein of the endoplasmic reticulum, had its catalytic site oriented towards the lumen of this organelle. Moreover, it showed that it had to be assisted by a glucose-6-phosphate transporter. This subsequently led to the recognition that GSD1b was due to a defect in this transporter [5,6].

As GSD1a and GSD1b became established as two separate defects, it became clear that GSD1b patients also suffered from neutropenia associated with neutrophil dysfuncsation in addition to the clinical symptoms found in GSD1a [7,8]. Therefore, the two diseases provided an informative system, based on which researchers tried to identify the cause of the neutrophil dysfunction. Yet, several major discoveries were needed before the origin of this neutropenia could be recently understood. First, Lei and co-authors [9] identified *G6PC1* as the gene encoding glucose-6-phosphatase, others identified *G6PC3* as the gene encoding a ubiquitously expressed glucose-6-phosphatase homolog, G6PC3 [10,11,12] and Van Schaftingen and co-authors [13,14] identified *SLC37A4* as the gene mutated in GSD1b patients encoding an also ubiquitously expressed glucose-6-phosphate transporter (G6PT). Next, a deficiency in G6PC3 was shown to lead to neutropenia in mice [15], and mutations in G6PC3 explained a still unknown form of severe congenital neutropenia [16] (Figure 1). Despite this progress, it was only after Veiga-da-Cunha, Van Schaftingen and their collaborators in a laboratory in Brussels elucidated the importance of metabolite repair for the efficient running of intermediary metabolism in cells [17,18] that the mechanism of neutropenia in GSD1b was finally understood.

Metabolite repair is a major process involving numerous metabolite repair enzymes that serve to eliminate toxic metabolites formed by enzyme side reactions. When applied to the neutropenia in GSD1b and G6PC3 deficiency, this concept led to the discovery that the molecule that intoxicates neutrophils was 1,5-anhydroglucitol-6-phosphate (1,5-AG6P), a compound without any known function that is structurally similar to glucose-6-phosphate and a known inhibitor of hexokinase 1, which is normally kept at a very low concentration by the combined activity of two proteins endowed with metabolite repair activity: G6PT and G6PC3 [17] (Figure 2). This eventually resulted in a successful treatment based on the lowering of 1,5-anhydroglucitol (1,5-AG) concentrations in blood with inhibitors of SGLT2, the renal Na^+^-glucose transporter in the proximal tubules [19]. We now know that SGLT2 inhibitors, by raising the concentration of glucose in the renal filtrate, indirectly inhibit the 1,5-AG transporter, which was only recently identified as SGLT5 [20,21]. 

Despite being quite a complex problem, we believe that it is important for those that are involved in treating these patients to understand the principles on which the treatment is based, because this is probably the best way to use it optimally in order to avoid side-effects and to maximize the benefit for patients. Consequently, this review starts by describing the enzymes and transporters involved in the pathophysiological mechanism of the neutropenias present in GSD1b and G6PC3 deficiency before reviewing the successful trials with gliflozins that have been published or are ongoing. It ends with some considerations on avenues that could still lead to further treatment improvement. 

## 2. The Enzymes

### 2.1. Glucose-6-Phosphatase (G6PC1)

Most of the glucose that is present in blood during fasting periods is formed from glycogen that is mostly stored in liver but also from gluconeogenic precursors both in the liver and kidney. During this course of action, both gluconeogenesis and glycogen breakdown lead to the production of glucose-6-phosphate in the cytosol of cells present in these organs. The enzyme that hydrolyses glucose-6-phosphate is glucose-6-phosphatase, an enzyme associated with the membrane of the endoplasmic reticulum. Glucose-6-phosphatase, often called G6PC1 because it is encoded by the *G6PC1* gene, belongs to a family of phosphatases that use histidine as an intermediary phosphoryl acceptor during the catalytic cycle [4,12]. G6PC1 is highly expressed in the liver and in kidneys, but it is also, to a lesser extent, expressed in the intestinal mucosa, where it is involved in the conversion of fructose to glucose [22,23]. 

Measurements performed in intact liver microsomes showed that G6PC1 is quite specific for glucose-6-phosphate. This is not true when microsomes are treated with detergents and G6PC1 becomes able to hydrolyze mannose-6-phosphate, the C2 epimer, at the same rate as glucose-6-phosphate [24]. This difference is due to the fact that glucose-6-phosphate is transported across the membrane of the endoplasmic reticulum specifically by a transporter of glucose-6-phosphate, often called G6PT, that is unable to transport mannose-6-phosphate. It is the specificity of this transporter against mannose-6-phosphate that prevents the formation of large amounts of free mannose in vivo.

### 2.2. The Glucose-6-Phosphate Transporter (G6PT)

Glucose-6-phosphate transporter (G6PT) is also a transmembrane protein. It is encoded by the *SLC37A4* gene, and it belongs to a family of transporters for different phosphate esters present in prokaryotes and eukaryotes [25,26]. The human genome also encodes three other members of the SLC37 family, but their function has not been precisely determined [27,28]. G6PT is also able to transport 1,5-AG6P (also called 1-deoxyglucose-6-phosphate), indicating that the presence of an OH group on C1 does not impact its specificity, contrary to the orientation of the OH group on C2 of the hexose-6-phosphate, as explained above for mannose-6-phosphate [17]. 

Contrary to G6PC1, G6PT has a wide tissue distribution, including neutrophils [13,29,30]. This is due to the fact that it serves to provide not only G6PC1 with substrates, with a restricted substrate specificity, but also two other enzymes with a wider tissue distribution: hexose-6-phosphate dehydrogenase and G6PC3. 

Hexose-6-phosphate dehydrogenase is a bifunctional enzyme that catalyzes two consecutive reactions in the endoplasmic reticulum: the NADP-dependent conversion of glucose-6-phosphate to 6-phosphogluconolactone and the irreversible hydrolysis of this lactone to 6-phosphogluconate. The function of this bifunctional enzyme is to produce NADPH in the lumen of the endoplasmic reticulum [31,32]. The latter serves for feeding endoplasmic reticulum reductases, particularly 11β-hydroxysteroid dehydrogenases 1 (11β-HSD1), which serves for the reduction of inactive, 11-keto forms of glucocorticoids (cortisone and 11-dehydrocorticosterone) to active, 11β-hydroxyl forms (cortisol and corticosterone) [33,34]. 

### 2.3. The ‘Ubiquitous Glucose-6-Phosphatase’ G6PC3 

The human genome encodes two proteins sharing sequence identity with G6PC1: 50% for G6PC2 [35] and 35% for G6PC3 [36]. Both proteins are also membrane proteins associated with the endoplasmic reticulum and are similarly oriented with the catalytic site facing the lumen [37]. G6PC2 is exclusively expressed in β-cells of pancreatic islets and serves as a mechanism to sensitize glucose metabolism and insulin secretion to changes in the blood glucose concentration [38]. Unlike G6PC1 and G6PC2 that have tissue-specific expressions, G6PC3 has a wide tissue distribution, including neutrophils, and a much broader substrate specificity than G6PC1 (when tested in the presence of detergents to avoid any limiting effect of substrate entry). It acts on a wide variety of phosphoric esters such as ribose-5-phosphate, glycerol-3-phosphate, ribitol-5-phosphate, mannose-6-phosphate and 1,5-AG6P, all better than on glucose-6-phosphate, which is a rather poor substrate [17]. Of note is the fact that to be truly physiological substrates for G6PC3, these compounds have either to be generated inside the endoplasmic reticulum (which is unlikely for all of them) or have to be transported from the cytosol, where they are generated, into the endoplasmic reticulum. In practice, the only established physiological substrate is 1,5-AG6P, which is transported (like glucose-6-phosphate) by G6PT. The structural difference between glucose-6-phosphate and 1,5-AG6P (also called 1-deoxyglucose-6-phosphate) is only minimal [17]. 

## 3. The Diseases

### 3.1. Glucose-6-Phosphatase (G6PC1) Deficiency or GSD1a

Glucose-6-phosphatase deficiency or von Gierke’s disease is due to inactivating mutations in the G6PC1 gene [9]. It is characterized by recurrent hypoglycemia, lactic acidosis, hypertriglyceridemia and glycogen accumulation in the liver. All of these are the consequence of the metabolic block, due to which glucose-6-phosphate can no longer be converted to glucose in the liver and kidneys. Neutropenia and neutrophil dysfunction do not belong to the clinical picture, though extremely rare cases of the association of G6PC1 deficiency with neutropenia have been described [39]. 

In a study aimed at determining whether there is a genetic or serologic predisposition of GSD1a patients (asymptomatic for inflammatory bowel disease—IBD) to develop IBD, the authors found that this was the case for a significant proportion of patients with GSD1a (11 out of 50 patients enrolled in the study) [40]. IBD in GSD1a patients, however, does not appear to be linked to neutrophil dysfunction, and the underlying mechanism remains unclear at present. A high-cornstarch diet, which favors the development of fermenting bacteria in the intestine; the possible role of intestinal G6PC1 in the metabolism of an unidentified compound; and a frequent association of G6PC1 mutation with mutations in an IBD gene due to a population bias have been evoked. Further work is needed to solve this question. 

### 3.2. Glucose-6-Phosphate Transpsorter (G6PT) Deficiency or GSD1b

Deficiency in G6PT gives rise to a similar metabolic phenotype as in GSD1a, to which neutropenia and neutrophil dysfunction are added. Consequently, G6PT deficiency results in various metabolic abnormalities in white blood cells, particularly in neutrophils and monocyte-derived macrophages, while lymphocytes appear to be unaffected or at least less affected [41,42]. As a consequence, GSD1b patients suffer from recurrent bacterial infections (specially skin, perianal and genital abscesses, upper respiratory track and intestinal infections) and painful mucosal lesions including oral and anogenital ulcers. The majority of GSD1b patients develop a Crohn-like inflammatory bowel disease (IBD) with diarrhea and chronical abdominal pain [43,44]. 

Until recently, none of the available therapeutic options have addressed the origin of the neutropenia and neutrophil dysfunction in GSD1b. Most patients took regular (and painful) injections of granulocyte-colony-stimulating factor (GCSF), a cytokine that is used to stimulate the proliferation and differentiation of neutrophil precursors in bone marrow. The stimulation of the bone marrow increases blood neutrophil counts but is associated with increased risk of development of myelodysplasia or acute myeloid leukemia [45,46]. Yet, despite this risk, and its limited efficacy in many patients, GCSF was and still is regularly used as a way to treat neutropenia in GSD1b and G6PC3-deficient patients. Of note, severe IBD in these patients has sometimes required invasive surgical treatment [47] and allogeneic haematopoietic stem cell transplantation (HSCT), which is associated with a high risk and is far from an optimal solution has been used to treat some G6PC3-deficient patients [48]. 

Since patients with GSD1a and GSD1b cannot convert glycogen or gluconeogenic precursors to glucose (by dephosphorylating glucose-6-phosphate in the endoplasmic reticulum), they usually take regular meals of uncooked cornstarch or a similar source of slowly digestible carbohydrates in order to prevent hypoglycemic episodes during fasting. Starches are important contributors to the high level of 1,5-AG that are found in these patients’ blood compared to healthy controls or even G6PC3-deficient patients. This is not a problem for GSD1a patients because they have a functional G6PT, but it further harms the neutrophil function in GSD1b patients. Consequently, by addressing their metabolic problem and preventing hypoglycemia, their neutropenia is further harmed, thus explaining why the recent understanding of this mechanism [17] and the identification of a treatment that reduces blood 1,5-AG [19], the precursor of toxic 1,5-AG6P in neutrophils, is such a breakthrough in the treatment of neutropenia in GSD1b patients. 

GSD1b and G6PC3-deficient patients sometimes show thrombocytopenia and often mild or severe anemia [49,50]. These problems might also result from 1,5-AG6P accumulation in the case of low platelet counts and of IBD-related intestinal iron malabsorption in the case of anemia. Indeed, following treatment with SGLT2 inhibitors, at least when anemia was an issue before treatment, it was often resolved following the enhancement of 1,5-AG urinary excretion [19,51,52,53,54,55,56]. This was less clear for thrombocyte counts [57].

### 3.3. G6PC3 Deficiency or Dursum Syndrome

G6PC3 deficiency, both in mice and in humans, does not give rise to a metabolic pathology, which is in agreement with the enzymological data indicating that the function of G6PC3 is not to dephosphorylate glucose-6-phosphate [17], as is the case for G6PC1 and G6PC2. Accordingly, in liver, the presence of a specific phosphatase (G6PC1) that dephosphorylates glucose-6-phosphate to produce glucose is meaningful, while in neutrophils, this is meaningless; why produce glucose from glucose-6-phosphate in neutrophils if the primary role of glucose is to feed glycolysis in these cells? It was this line of thought that led Veiga-da-Cunha and Van Schaftingen to question the function of G6PC3 as a glucose-6-phosphatase and eventually resulted in the discovery of its physiologically essential 1,5-AG6P phosphatase activity in granulocytes. The phenotype in G6PC3-deficient patients comprises a consistent problem of neutropenia and neutrophil dysfunction, in addition to the presence of a typical prominent superficial venous pattern clearly seen on the torso and legs, as well as cardiac malformations that often need surgical correction and urogenital malformations [58,59]. 

As for GSD1b patients, G6PC3-deficient neutrophils have severe defects of the glucose metabolism that result in neutropenia and neutrophil dysfunction [60], which are associated with the same susceptibility to infections as seen in GSD1b patients. The origin of the neutrophil problem in G6PC3-deficient neutrophils is now understood as being the consequence of 1,5-AG6P accumulation [17], while the link between deficient G6PC3 activity and the presence of the characteristic superficial venous pattern, as well as the cardiac and urogenital malformations, still remains unclear. Since these malformations are not present in GSD1b patients, this indicates that they are likely unrelated to 1,5-AG6P accumulation. Though G6PC3 deficiency is metabolically unrelated to GSDs 1a and 1b, its neutropenia/neutrophil dysfunction is pathophysiologically closely related to the one found in GSD1b and can be treated with gliflozins in the same manner as GSD1b-related neutropenia [19,20], thus justifying its description in the present review (see Figure 1 and Figure 2). 

## 4. Pathophysiological Mechanisms Underlying the Neutropenia

### 4.1. Diseases of Metabolite Repair

Unlike what is usually stated in textbooks of biochemistry, enzymes of intermediary metabolism are not sufficiently specific to avoid the formation of side products. Side products are formed, but many of them are destroyed by a series of enzymes acting as “metabolite repair enzymes”. An example is L-2-hydroxyglutarate dehydrogenase (L-2-H-GDH), an enzyme that serves to oxidize a side product of the activity of L-malate dehydrogenases and L-lactate dehydrogenases on α-ketoglutarate [61]. L-2-hydroxyglurate has no function, but it accumulates in tissues, particularly the brain, when the repair enzyme L-2-H-GDH is inactivated by mutations. This results in a severe progressive neurological disease [62,63]. Other examples of inborn errors due to defective metabolite repair are the defects in the enzymes repairing damaged forms of NAD(P)H or deaminated glutathione that have been recently reviewed [18,64].

### 4.2. 1,5-. Anhydroglucitol-6-Phosphate (1,5-AG6P) Accumulation

The neutropenia in GSD1b and in G6PC3 deficiency results from the accumulation of 1,5-AG6P, an inhibitor of low-Km hexokinases. 1,5-AG6P is formed by a side activity of hexokinases (mainly hexokinase 3 and 1, the two most abundant forms in neutrophils) and of ADP-glucokinase (ADPGK), which normally serves to phosphorylate glucose but also acts on the 1,5-AG that is present in blood, tissues and cells [17]. In cells, the concentration of 1,5-AG6P is normally kept low, allowing hexokinase to phosphorylate glucose as a result of the activity of G6PT and of G6PC3, which collaborate to dephosphorylate 1,5-AG6P in a metabolite repair function (see Figure 2). When 1,5-AG6P is allowed to accumulate in neutrophils, it inhibits hexokinase by binding probably to both the inhibitory allosteric site for glucose-6-phosphate (in the N-terminal domain) as well as to the site in the catalytically active C-terminal domain responsible for the competitive inhibition of hexokinases by glucose-6-phosphate [17,65,66].

Hexokinases play an essential role in the energy metabolism in neutrophils because these cells rely essentially on glycolysis for the provision of ATP owing to the fact that, during neutrophil differentiation in the bone marrow, glucose metabolism (and ATP production) is shifted from oxidative phosphorylation in mitochondria to glycolysis with a loss of cytochrome c expression and functional mitochondria in general [67,68]. As a result, in circulating neutrophils, mitochondria do not play a role in ATP production [69,70], and neutrophils rely on glycolysis for energy production. Hexokinase activity is also essential for providing the glucose-6-phosphate needed for NADPH production, which is essential for the respiratory burst and the provision of reactive oxygen species that contribute to the killing of bacteria. It is also key for the production of UDP-glucose for glycogen synthesis and glycans required for the glycosylation of proteins [71]. 

Consequently, the proposed mechanism (inhibition of hexokinase) is in full agreement with data indicating that neutrophils in GSD1b and in G6PC3 deficiency have a lower rate of glucose metabolism, have a reduced motility and viability, have a reduced respiratory burst and show a defective glycosylation of proteins [71]. Together, this likely also explains the increased endoplasmic reticulum stress and ROS production mentioned in GSD1b and in G6PC3-deficient neutrophils and macrophages [72]. These features are corrected by lowering the concentration of 1,5-AG in the blood, thereby reducing the intracellular concentration of 1,5-AG6P [17,19]. This further supports the proposed pathophysiological mechanism.

Of note is the fact that the previously proposed and still often-cited function of G6PC3, which was to dephosphorylate glucose-6-phosphate in order to provide more glucose to glycolysis [73], has a major interpretation problem. If, as previously suggested, the physiological function of G6PC3 was to hydrolyze glucose-6-phosphate in neutrophils, this would remove an essential glycolytic intermediate for neutrophils to function. Moreover, if this was the case, G6PT or G6PC3 deficiency would be expected to increase glucose-6-phosphate (and not reduce it) and therefore increase flux through glycolysis and the pentose–phosphate pathway, and exactly the opposite was observed [17].

### 4.3. 1,5-. Anhydroglucitol (1,5-AG)

In view of the crucial role that 1,5-AG plays in GSD1b and G6PC3-deficient neutropenias, it is important to understand its origin, its fate and its mechanism of excretion. Curiously, 1,5-AG does not seem to have a physiological role in humans and other vertebrates, yet it is present in blood at a concentration of about 150 µM in healthy individuals [74,75].

1,5-AG is present in food, and, according to studies performed about 40 years ago, food is by far its principal source, with endogenous formation representing only about 10% [76,77]. It derives from the depolymerization of polysaccharides by a lyase reaction, which produces 1,5-anhydrofructose rather than the much more common hydrolysis reaction, which produces glucose [78,79]. 1,5-Anhydrofructose is readily reduced by an intracellular enzyme called 1,5-anhydrofructose reductase, which produces 1,5-AG [80]. This sequence of reactions accounts for the formation of 1,5-AG not only in our body but also in our food, explaining the wide presence of this polyol in food, rendering the design of a 1,5-AG-free diet nearly impossible. It is therefore preferable to act on its excretion if one wishes to decrease its concentration in blood. 

1,5-AG enters cells most likely via the same transporters used by glucose. It is only very slowly and very partially phosphorylated to 1,5-AG6P [81], which, as mentioned above, is efficiently dephosphorylated [17]. There is no known further metabolism of 1,5-AG [82], which exits cells most likely via the same passive transporter that allowed its entry. 

What is more interesting to understand is the mechanism of its excretion, because 1,5-AG is efficiently reabsorbed in kidney tubule [83]. This results in a half-life of 1,5-AG in the body of more than 1 month [77]. The renal reuptake is carried out by the Na^+^-dependent transporter SGLT5 (encoded by the gene SLC5A10) [20,21]. This transporter belongs to the same family as the Na^+^-dependent glucose transporters SGLT1 (present mainly in the intestine) and SGLT2, the main Na^+^-dependent glucose transporter in the kidney proximal tubule [84]. It appears likely that the physiological role of SGLT5 is to reabsorb mannose and fructose, two important sugars that share structural similarity with 1,5-AG [85,86]. 

SGLT5 is competitively inhibited by glucose [21], so that the reabsorption of 1,5-AG is inhibited in uncontrolled diabetes or in (diabetic) patients treated with SGLT2 inhibitors known as gliflozins. These two conditions are associated with decreased 1,5-AG concentrations in blood [87,88]. 

It is the increase in the concentration of glucose in the proximal tubules, caused by SGLT2 inhibitors, that indirectly inhibits (by competition) the renal reabsorption of 1,5-AG by SGLT5 [89]. This mechanism is the basis for the treatment of the neutropenia associated with G6PC3 and G6PT deficiency. Moreover, SGLT5 also shows some sensitivity to a direct inhibition by SGLT2 inhibitors themselves, which is roughly 1000-fold lower compared to the inhibition of SGLT2 by gliflozins [21,85,90]. This indicates that a possible (likely smaller) direct effect of SGLT2 inhibitors on SGLT5 might also contribute to the observed 1,5-AG-lowering effect (Figure 3).

## 5. Role of the Neutrophil Dysfunction and Other Aspects of Immune Dysregulation in the Pathophysiology of the Infections

The neutropenia observed in G6PC3 and G6PT deficiency certainly plays a role in the development of infections. However, it is likely that neutrophil dysfunction is more important than low neutrophil counts (neutropenia) in this process. It is striking that IBD is not frequently observed as a consequence of other genetic forms of neutropenia. For example, IBD does not seem to be common in the severe congenital neutropenia caused by mutations in ELA2, CSFR3 or HaX1 [91]. On the contrary, it is frequently observed in chronic granulomatous diseases, which are not characterized by low neutrophil counts (neutropenia) but by the incapacity of phagocytes (including neutrophils and macrophages) to kill certain types of bacteria and fungi. This results from a deficiency in the NADPH–oxidase complex due to mutations in one of the four essential subunits forming the enzymatic complex required for activity [92]. 

NADPH–oxidase produces reactive oxygen species (ROS) that are essential not only for neutrophil but also for macrophages to kill bacteria. Of note is the fact that for NADPH oxidase to function it requires the production of NADPH, which in neutrophils and macrophages is essentially made by the enzymes catalyzing the two NADPH-producing steps of the pentose phosphate pathway. Consequently, NADPH supply is compromised in neutrophils (and most likely also macrophages) from G6PC3- and G6PT-deficient subjects. In agreement, Chou and collaborators [93] have reported that macrophages isolated from a G6PC3-deficient mouse model have a lower glucose uptake and lower levels of glucose-6-phosphate, lactate and ATP compared to macrophages isolated from wild-type mice. Moreover, these macrophages also show a reduced respiratory burst. Since these macrophages were isolated from G6PC3-deficient mice, it is likely that they were loaded with 1,5-AG6P and that the inhibition of hexokinase played an important role in this dysfunction, as was previously found for neutrophils from the same mouse model [17].

This indicates that the defect in the production of ROS (and therefore the neutrophil dysfunction), rather than the neutropenia per se, is probably a major contributor to the IBD present in G6PC3 or G6PT deficiency. This may explain the limited efficacy of GCSF therapy on the improvement of IBD in these patients [19,45,50,94,95], as GCSF mainly results in an increased neutrophil count but a minimal effect on functionality, particularly in terms of improving the neutrophils’ ability to perform a respiratory burst. 

Some authors have recently questioned a possible role for other cells of the immune system in the susceptibility to infection and particularly to inflammation found in these patients. Accordingly, Melis and collaborators [96] reported lymphopenia in GSD1b patients with a reduction in natural killer (NK) cells, CD4^+^T cells and CD8^+^T cells, which was absent in GSD1a patients and controls. They also described a reduced capacity of T lymphocytes to engage in glycolysis upon stimulation. The origin of this observation remains unexplained, but it is possibly also caused by the toxic accumulation of 1,5-AG6P in certain subpopulations of peripheral blood mononuclear cells (PBMCs). Accordingly, Veiga-da-Cunha and collaborators [20] observed that isolated PBMCs from a G6PC3-deficient patient also accumulated 1,5-AG6P (although 10-fold lower than in neutrophils) and also appeared to have a suboptimal glycosylation of the hyperglycosylated protein LAMP2 in PBMCs, which was corrected following treatment with SGLT2 inhibitors. This observation, together with the results from Melis and collaborators [96], suggests that the accumulation of 1,5-AG6P in white blood cells might go beyond just neutropenia and neutrophil dysfunction. In this case, the treatment with SGLT2 inhibitors would benefit all cell populations that accumulate 1,5-AG6P in GSD1b and G6PC3-deficient patients.

## 6. Treatment with SGLT2 Inhibitors (Gliflozins)

### 6.1. Preliminary Note 

The treatment of neutropenia and neutrophil dysfunction in GSD1b and G6PC3-deficient patients has not benefited from a classical clinical trial where, for example, the effects of GCSF and empagliflozin are directly compared in two groups of patients. This is ethically justified because the repurposing of empagliflozin to treat the neutropenia in both diseases has clearly improved neutrophil counts, neutrophil dysfunction and the severity and number of infectious episodes, while this was often not the case under GCSF treatment. This is not surprising, because while the use of gliflozins directly addresses the cause of the neutropenia by decreasing the accumulation of the toxic 1,5-AG6P in the patients’ neutrophils, GCSF mainly stimulates the production of neutrophils in the bone marrow, but these will still accumulate 1,5-AG6P and therefore remain largely dysfunctional. Yet, the use of gliflozins in the treatment of neutropenia in these patients needs to be considered as experimental, and SGLT2 inhibitors have to be used with great caution and always under careful medical supervision. Moreover, it is important to continue to report any adverse effects as well as successful outcomes to a central registry in order to allow all the patients and clinicians to benefit from these observations.

To our knowledge, there are now sixteen published reports on treatment with SGLT2 inhibitors in both GSD1b [19,51,52,53,54,55,56,57,97,98,99,100] and G6PC3-deficient [20,95,101,102] patients where empagliflozin was repurposed to treat these patients’ neutropenia. One of these studies assembled clinical data collected through a questionnaire of 112 treated individuals with GSD1b from 24 different countries [103].

### 6.2. Indication 

GCSF has been used as a standard treatment for the neutropenia and neutrophil dysfunction in GSD1b patients since the beginning of the 1990s [104,105]. This cytokine is usually injected on a regular basis or in some cases only during infection since it increases neutrophil counts in blood within a few hours. In a review of 18 European GSD1b patients that were given GCSF, apart from hematological effects (increase in neutrophil counts), there was little unequivocal improvement in the clinical outcome, even if the occurrence of infections was lower and IBD was reduced subjectively, though this was not the case in all patients [50,106]. This suggests that GCSF treatment is not satisfactory. Moreover, the daily injections are painful (particularly for young children), they do not prevent the need for recurrent hospitalizations, they are an important cause of osteopenia, they do not prevent the iron deficiency and anemia nor the thrombocytopenia (when seen), they are often accompanied by serious splenomegaly and they increase the risk of blood cell malignancy [45,50]. 

It is against this background that, following the pioneering study published by Wortmann, Veiga-da-Cunha and collaborators [19], as mentioned above, many clinicians have opted for the off-target use of empagliflozin to treat the neutropenia in their GSD1b and G6PC3-deficient patients. From the largest published study on GSD1b patients [103], the following benefits of the treatment are indeed quite impressive: an improvement in the neutropenia with a three-fold reduction in the severe forms (ANC <500/μL, from 29% to 10%); a marked increase (17% to 53%) in the number of patients with no neutropenia; and an important decrease in recurrent oral/anogenital mucosal lesions (from 68% to 13%), in recurrent bacterial/skin infections (from 54% to 8%), in the severe and moderate forms of IBD (from 37 to 6%) and in anemia (from 73% to 30%). Importantly, following empagliflozin therapy, GCSF injections could be discontinued in more than 50% of the patients that required it before the initiation of empagliflozin (from 84% and 37%) and could be reduced either in dose or the frequency of injections in about 40% of the other patients. No mention was made of any definitive interruption of empagliflozin, suggesting a high degree of satisfaction. 

In the present state of knowledge, treatment with empagliflozin (or another gliflozin) has therefore to be taken into consideration for most of the neutropenic GSD1b (and G6PC3-deficient) patients, certainly those that have chronic treatment with GCSF and for which hospitalizations due to infections continue to be frequently needed. The exception would be the mildest forms of neutropenia, for which no treatment with GCSF is needed. In such cases, the potential improvement in general well-being and quality of life has to be taken into consideration. 

### 6.3. Dosage, Contraindications and Adverse Effects 

Contraindications and adverse effects have mainly been listed in the study by Grünert et al. [103]. Accordingly, the main reasons to avoid starting to treat neutropenia in GSD1b patients with an SGLT2 inhibitor is probably the presence of recurrent hypoglycemias linked to suboptimal metabolic control. In this case, the diet should be optimized in order to achieve a stable metabolic condition before starting empagliflozin treatment. 

Kidney function also needs to be checked. In type-2 diabetes, for which empagliflozin treatment is approved, the current U.S. Food and Drug Administration prescribing information allows for empagliflozin to be used in adult patients with an eGFR ≥ 30 mL/min/1.73 m^2^ [107]. These recommendations concern an adult population of type-2 diabetics. For children and adolescents, there are no available guidelines yet, but a few of the SGLT2 inhibitors that are currently available in the market are undergoing phase-three clinical trials in this population. In GSD1b and G6PC3-deficient patients, SGLT2 inhibitors can only be used as an off-label treatment so far. In the meantime, several infants with GSD1b have already been successfully treated. To our knowledge, adverse effects specifically linked to its use in young patients have not been reported, indicating that treatment within the doses that have been described also appears to be safe for this population. The median dose used for empagliflozin in the published cases was 0.35 mg/kg/day (0.3 in adults and 0.4 in children), ranging in all patients from 0.1 to 0.9 mg/kg/day [103].

When treatment was initiated, the most common adverse effect reported was level three hypoglycemia, defined as <3.0 mmol/L (54 mg/dL) and characterized by an altered mental and/or physical status requiring assistance (18% of all patients—20/111) [103]. Along the same lines, Halligan and colleagues [54] observed hypoglycemic episodes in 50% of their cohort of eight GSD1b patients following the commencement of empagliflozin therapy. These were often solved by adapting the daily cornstarch intake, which the authors suggest could be lower in their center, compared to patients enrolled in other published studies where this adverse effect has not been reported [19,51,52,53,56,57,97,98,99].

A list of other side effects, among those listed in the drug information and collected in a study including 112 GSD1b patients, comprises an allergic/anaphylactic reaction (1%), fungal urogenital infections (3%), urinary tract infection (7%), skin rash (3%), puritis (1%), ketoacidosis (1%), lactic acidosis (5%) and dehydration (1%) [103]. Since empagliflozin is associated with a risk of ketoacidosis, it is advisable to pause the administration of empagliflozin during severe keto and lactic acidosis as well as dehydration, which may appear in the setting of diarrhea associated with gastrointestinal infections [103]. Once the episode is resolved, the treatment may be restarted. Of note is the fact that this short interruption of the gliflozin treatment should only lead to a very slow increase in the blood concentration of 1,5-AG, since once 1,5-AG has been depleted it takes at least 3 months to re-establish the body pool of 1,5-AG [77].

### 6.4. Choice of the Gliflozin 

In principle, all gliflozins should be suitable to be used in the treatment of neutropenia and neutrophil dysfunction in GSD1b and G6PC3-deficient patients. Yet, in most of the reports available, empagliflozin was the one chosen, as was already extensively referenced above. However, the authors are aware of a large study presently being conducted in France [108], where dapagliflozin is being tested in GSD1b and G6PC3-deficient patients, and the results indicate similar therapeutical success once the dose has been adapted. Because of the limited number of patients (ultra-rare disease), it might be better to choose gliflozins that have already been tested (if available) to gain better knowledge of their efficacy and secondary effects. 

It is, however, possible that in the future, a gliflozin with a lower degree of specificity for SGLT2, such as remogliflozin [90], might be of interest to test. Remogliflozin is not commercialized in Europe, but it is well commercialized in India [109]. It is generally admitted that the effect of gliflozins on 1,5-AG excretion is indirect and caused by the enhanced glucosuria. In addition, recent work has identified SGLT5 as the main 1,5-AG transporter in the kidney [21] and has shown that gliflozins can also directly inhibit the transport of 1,5-AG by SGLT5, suggesting the possibility that part of the effect of gliflozins on 1,5-AG excretion that is seen in vivo could be due to a direct effect on SGLT5. In this context, remogliflozin has been shown to better inhibit the transport of 1,5-AG in model cell lines overexpressing SGLT5 compared with empagliflozin and dapagliflozin [21]. Since the inhibition constants obtained are in the range of the concentration of empagliflozin and remogliflozin reported in the plasma of treated patients taking these drugs [110,111], it would be interesting to compare the effect of these two gliflozins in vivo. Clinicians might therefore like to test whether remogliflozin could have a stronger effect on promoting the urinary excretion of 1,5-AG and further lowering its concentration in blood. This could be particularly helpful for GSD1b patients that have higher blood 1,5-AG concentrations due to their starch-based diet, and it may possibly allow more patients to finally stop GCSF treatment when taking SGLT2 inhibitors. 

### 6.5. Effect of Mutations in SGLT5

Mutations of SGLT5 have been associated with lower levels of 1,5-AG in the general population [112,113,114] and in at least one patient with G6PC3 deficiency [20]. In addition, Veiga-da-Cunha and collaborators have also shown that these mutations in SGLT5 do indeed decrease the ability of SGLT5 to transport 1,5-AG in model cell lines that overexpress the mutant transporters [21]. Since SGLT5 is the renal transporter for 1,5-AG and these mutations in the heterozygous state show a decrease in the concentration of blood 1,5-AG of 50%, we conclude that for GSD1b and G6PC3-deficient patients, it can be an advantage to carry at least one of the SGLT5-mutated alleles. This was indeed the case for a G6PC3-deficient patient with a heterozygous mutation in SGLT5 (Arg401His) [20] that reduced the transport activity of 1,5-AG by 50% [21]. This patient not only had a milder form of neutropenia before the initiation of empagliflozin treatment that only occasionally required GCSF injections but also had an impressive response to treatment that resulted in a perfect normalization of the ANC (4000/µL) with a particularly low dose of empagliflozin (0.1 mg/kg/day). As such, according to the frequency of SGLT5 inactivating sequence variations in various European, African and South Asian populations (see [21]), it is predicted that among the GSD1b and G6PC3-deficient population, 2 to 3% of them are likely to have a defective renal re-uptake of 1,5-AG and will likely respond better to the SGLT2 therapy (and therefore need lower doses of empagliflozin) to treat their neutropenia.

### 6.6. Could SGLT2 Inhibitors Help GSD1b Patients beyond Treating Their Neutropenia?

A long-term consequence of GSD1b (and GSD1a) is kidney failure, which results from the progressive accumulation of glycogen in the kidneys that worsens under suboptimal metabolic control. As such, variable stages of kidney disease, from proximal and distal tubulopathies to irreversible glomerular injury, may affect the outcome of these patients [115]. In a mouse model of GSD1b that accumulated glycogen in the kidney, Trepiccione and collaborators [116] showed that the main protein markers of proximal tubule function were all down-regulated. These markers included the major sodium and phosphate transporters along the proximal tubule (NHE_3_ and NaPi2A), the two main renal glucose transporters located on the apical and the basolateral membranes of the proximal tubular cells (SGLT2 and GLUT2) and AQP1, the principal aquaporin water-transporting protein in the plasma membranes of the proximal tubular cells. When the GSD1b model mice were treated with the SGLT2 inhibitor dapagliflozin, the authors observed a beneficial effect on the reduction in glycogen accumulation in the proximal tubular cells, which they linked to a recovery in the expression of the main proximal tubule protein markers listed above, leading to improved renal function in the mice. 

Although performed in a GSD1b mouse model, this study could motivate clinicians treating GSD1b patients with gliflozins to carefully monitor kidney function in their patients to see if they are able to identify kidney-specific beneficial effects that go beyond treating neutropenia for GSD1b patients. 

## 7. Conclusions and Outlooks

Neutropenia and neutrophil dysfunction in GSD1b and G6PC3 deficiency are the result of a lack of specificity of various transporters as well as of the side activities of several enzymes. This results in the accumulation of 1,5-AG6P, a potent inhibitor of hexokinases that intoxicates neutrophils by blocking glucose metabolism (Figure 4).

In this context, the treatment of the neutropenia associated with GSD1b and G6PC3 deficiency with SGLT2 inhibitors offers a clear improvement compared to previous therapeutic approaches. However, it still has to be considered as experimental, and therefore careful monitoring is required. The understanding of the pathophysiology of these diseases suggests that there is still room for improvement, with better (and maybe specific) inhibitors of 1,5-AG reuptake or lowering the dietary content of 1,5-AG or its intestinal production being potential avenues for investigation.

Thus, it would be most interesting to understand why some patients respond better to gliflozin treatment than others. Is it linked to differences in 1,5-AG transport, phosphorylation in neutrophils or the quantity present in the food? These are questions that still need to be solved by the medical and scientific community and will require the active sharing of clinical data from patients undergoing off-label treatment. 

## Figures and Tables

**Figure 1 diagnostics-13-01803-f001:**
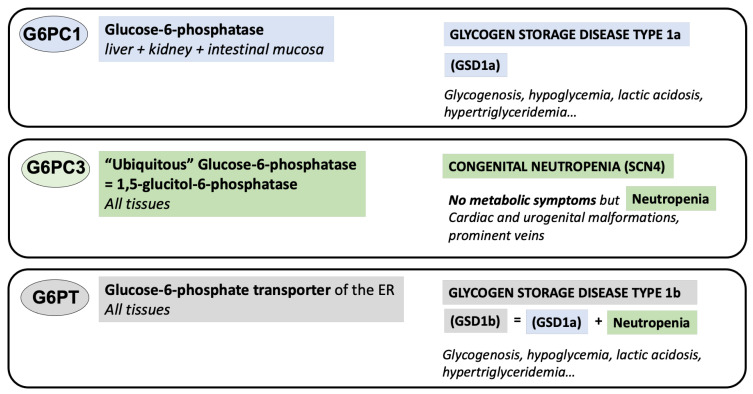
Diseases associated with G6PC1, G6PC3 and G6PT deficiencies.

**Figure 2 diagnostics-13-01803-f002:**
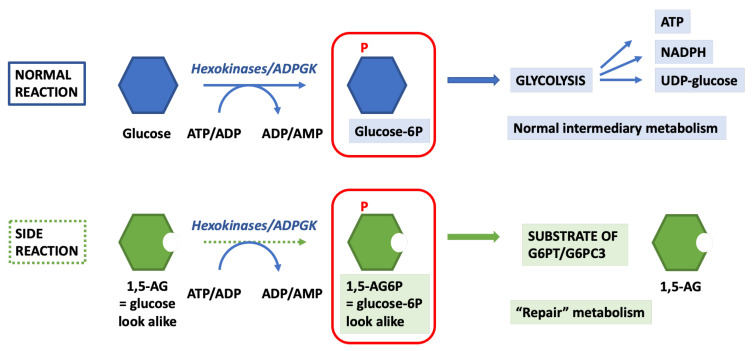
1,5-anhydroglucitol-6-phosphate (1,5-AG6P) is formed by side reactions of hexokinases and ADP-glucokinase (ADPGK), the glucose-phosphorylating enzymes in cells. Hexokinases and ADPGK phosphorylate glucose to glucose-6-phosphate, which feeds glycolysis to produce ATP required for cell survival, NADPH crucial for the respiratory burst reaction in neutrophils and UDP-glucose for glycogen synthesis and the production of glycans needed for protein glycosylation. The problem in neutrophils is that these glucose-phosphorylating enzymes also catalyze a side reaction on 1,5-anhydroglucitol (1,5-AG), the most abundant polyol in blood, which is structurally similar to glucose; they make 1,5-AG6P (structurally similar to glucose-6-phosphate), which is an inhibitor of hexokinases. To prevent its accumulation, G6PT and G6PC3 collaborate to dephosphorylate 1,5-AG6P back to 1,5-AG in a dead-end metabolite repair reaction.

**Figure 3 diagnostics-13-01803-f003:**
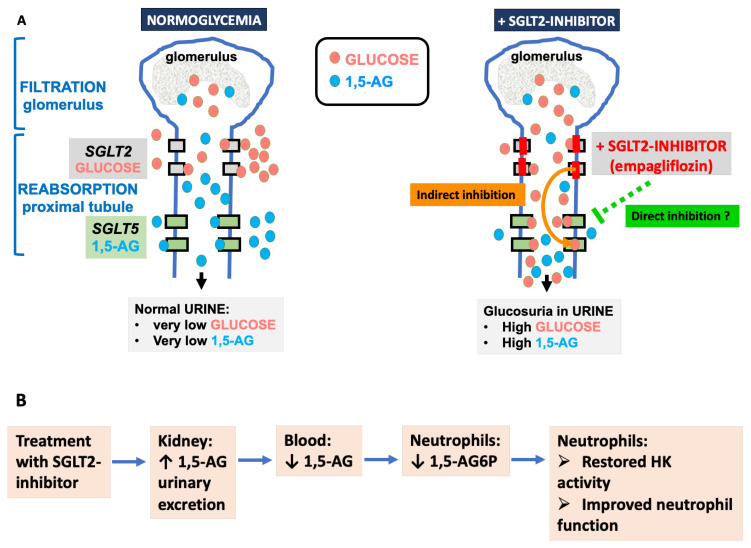
SGLT2 inhibitors indirectly and maybe also directly inhibit renal reabsorption of 1,5-AG, thus treating neutropenia in GSD1b and G6PC3 deficiency. (**A**) When blood is filtered in the kidney, glucose and 1,5-AG present in the renal filtrate are reabsorbed by SGLT2 and SGLT5, respectively, which prevent their urinary loss (left panel). In the presence of SGLT2 inhibitors, the concentration of glucose increases in the renal filtrate and indirectly inhibits the renal reabsorption of 1,5-AG by SGLT5. Recent results obtained in vitro [21] indicate that SGLT2 inhibitors may also directly inhibit SGLT5 in vivo. (**B**) As a result of SGLT2 inhibition, SGLT5 inhibition lowers renal reabsorption of 1,5-AG and favors its urinary excretion, which results in a decrease in the concentration of 1,5-AG in blood and 1,5-AG6P in neutrophils.

**Figure 4 diagnostics-13-01803-f004:**
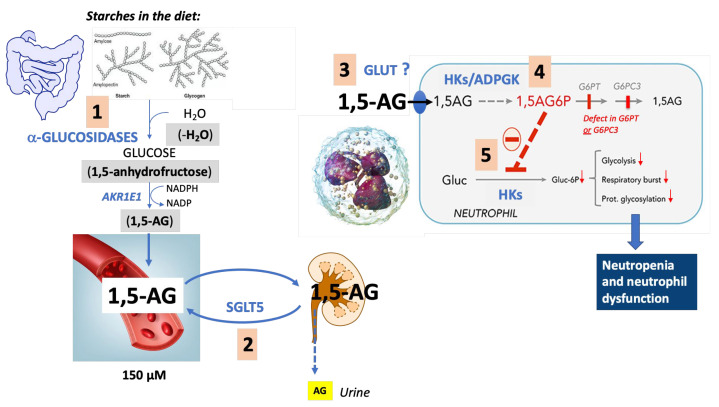
Lack of specificity of various enzymes and transporters is solved by a dedicated metabolite-repair system (G6PT and G6PC3) that prevents neutropenia. (1) 1,5-anhydroglucitol (1,5-AG) is present either in food or formed in the gut by a still-unknown process that may involve a side activity of intestinal (or bacterial) α-glucosidases. In this case, the digestion of starch may occasionally produce a molecule of 1,5-anhydrofructose (rather than of glucose) if, instead of using a water molecule in the breakdown of starch, α-glucosidases catalyze an elimination reaction. The 1,5-anhydrofructose molecule is quickly reduced to 1,5-AG by a ubiquitous NADPH-dependent 1,5-anhydrofructose reductase (AKR1E1—[80]). (2) Once transported across the intestinal wall, 1,5-AG stays in blood due to a lack of specificity of SGLT5, a mannose and fructose Na^+^-dependent transporter present in the kidney (proximal tubular cells) that also reabsorbs the polyol from the urinary filtrate, preventing its urinary excretion. (3) Probably as a result of its structural similarity to glucose, 1,5-AG can be mistakenly transported in cells by passive glucose transporters of the GLUT family and (4) phosphorylated by side activities of hexokinases (HKs) and ADP-glucokinase (ADPGK) to 1,5-anhydroglucitol-6-phosphate (1,5-AG6P). (5) In the neutrophils of GSD1b and G6PC3-deficient patients, 1,5-AG6P cannot be dephosphorylated and accumulates to concentrations that become able to bind to the glucose-6-phosphate inhibitory site of HK1 and HK3, inhibit the phosphorylation of glucose and thereby glycolysis, the pentose–phosphate pathway, NADPH production (respiratory burst) and the production of UDP-glucose (protein glycosylation), explaining the neutropenia and neutrophil dysfunction in GSD1b and G6PC3 deficiency [17].

## Data Availability

There were no new data created. This is a review of data published elsewhere.

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
