# Peer review of "Treatment of the Neutropenia Associated with GSD1b and G6PC3 Deficiency with SGLT2 Inhibitors"

_diagnostics, 2023, doi:10.3390/diagnostics13101803_

Round 1

Reviewer 1 Report

The manuscript of Maria Veiga-da-Cunha et al. is reviewing existing data on benefits of SGLT-2 in neutropenia associated with GSD1b and G6pC3 defficiencies. In general the manuscript is very detailed and thoroughly written. The concept is clearly described and accelerated elimnination 1,5-AG6P indeed may play a crucial role in inmpovement of patient's condition. But what for me is missing, is the effect of glucose elimination from the organism due to SGLT-2 inhibors. It is nkown that metabolism and immune function are closely related through maintenece of redox balance (PMID: 32033390 ). May be this can be shortly discussed in revised version of the manuscript in 1 or 2 paragraphs?

I would suggest also a scheme that will visualize effect of SGLT-2s on disease, so that reader can have simple and srtraightforward summary of what authors suggest as the effect of SGLT-2s on disease course. 

The manuscript of Maria Veiga-da-Cunha et al. is reviewing existing data on benefits of SGLT-2 in neutropenia associated with GSD1b and G6pC3 defficiencies. In general the manuscript is very detailed and thoroughly written. The concept is clearly described and accelerated elimnination 1,5-AG6P indeed may play a crucial role in inmpovement of patient's condition. But what for me is missing, is the effect of glucose elimination from the organism due to SGLT-2 inhibors. It is nkown that metabolism and immune function are closely related through maintenece of redox balance (PMID: 32033390 ). May be this can be shortly discussed in revised version of the manuscript in 1 or 2 paragraphs?

I would suggest also a scheme that will visualize effect of SGLT-2s on disease, so that reader can have simple and srtraightforward summary of what authors suggest as the effect of SGLT-2s on disease course. 

Author Response

Please see attachement answer reviewer 1

Reviewer 2 Report

The review “Treatment of the neutropenia associated with GSD1b and G6PC3 deficiency with SGLT2-inhibitors” by Veiga-da-Cunha et al is a thorough and balanced review which explains well the rationale behind the novel use of SGLT2 inhibitors as Gsd1b therapeutics and also provides a good biochemical background on the disease. Moreover, the review clarifies well why Gsd1b, and not Gsd1a, involves neutropenia. Neutropenia is caused by deficiency in G6PT, unique to Gsd1b, and also by G6PC3 deficiency, both of which causing accumulation of the hexokinase inhibitor 1,5-AG6P. The review also provides a thorough review of the clinical observations in Gsd1b and G6PC3 deficiency and their treatment by the SGLT2 inhibitors gliflozins. However some revisions and clarifications are still required:

1) “The enhanced urinary excretion of glucose inhibits the 1,5-AG transporter”. It is not clear on first read why there is enhanced glucose excretion. I guess this is due to inhibition of glucose re-absorption to the blood in SGLT2 inhibitor treated patients, but this needs to be spelled out.

2) Line 61 “G6PC3 (a phosphatase homologous to G6PC1)”. G6PC3 was already defined. There is no need to define it again and anyway the second definition is not specific enough. It has to be stated that this phosphatase is less specific to glucose-6-phosphate (and also dephosphorylates 1,5- glucitol-6-phosphate).

3) Line 62 – please correct to “unknown”

4) “When applied to the neutropenia in GSD1b and G6PC3 deficiency, this concept led to the discovery that the molecule that intoxicates neutrophils was 1,5-anhydroglucitol-6-phosphate (1,5-AG6P), a compound without any known function, which is normally kept at a very low concentration by the combined activity of two proteins endowed with metabolite repair activity: G6PT and G6PC3 [17] (FIG. 2).” – This sentence is not so clear on first read. The authors need to explain that 1,5-AG6P is toxic because it competes with G6P (at least this is my understanding).

5) Please add SGLT2 and its inhibition to Fig. 2.

6) Fig. 2: “UDP-glucose to make glycans needed for protein glycosylation.” UDP-glucose is also needed to synthesize glycogen.

7) Fig. 2: “they make 1,5-AG6P (structurally similar to glucose-6-phosphate), which is an inhibitor of hexokinases.” 1,5-AG6P also substitutes G6P leading to a dead end in all the reactions for which G6P is a precursor.

8) 1,5-AG6P is 1,5-anhydroglucitol-6-phosphate on page 2, but 1-deoxyglucose-6-phosphate on p. 4.  (1,5-AG6P)

9) Line 156: The word “only” is not needed.

10) Please spell out IBD (inflammatory bowel disease) on first mention.

11) p. 5: “Since patients with GSD1a and GSD1b cannot convert glycogen or gluconeogenic precursors to glucose.” This sentence is clear for gluconeogenic precursors, but for glycogen breakdown the authors need to explain how deficiency in dephosphorylation of glucose-6-phosphate affects glucose production from glycogen (i.e., via glucose-1-phosphate the main direct breakdown product of glycogen).

12) Line 254: “The neutropenia in GSD1b and in G6PC3 deficiency results from the accumulation of 1,5-AG6P, an inhibitor of low Km hexokinases.” Isn’t 1,5-AG a competitive inhibitor of hexokinase (Fig. 1)? Or do the authors refer to product inhibition by 1,5-AG6P?

13) Line 271: “It is also key for the production of UDP-glucose and glycans required for the glycosylation of proteins [69].” Again – please also add glycogen.

14) Line 303: “1,5-AG enters cells most likely via the same transporters as used by glucose. It is only very slowly and very partially phosphorylated to 1,5-AG6P [78] and, as mentioned above, efficiently dephosphorylated [17].” I guess the authors meant “which, as mentioned above, is efficiently dephosphorylated [17].”

15) Line 307: the mechanism

16) Line 360: “Since these macrophages were isolated from mice, it is likely that they are loaded with 1,5-AG6P and that inhibition of hexokinase plays an important role in this dysfunction, as previously found for neutrophils of G6PC3-deficient mice [17].” Unclear. Why should macrophages isolated from mice be loaded with 1,5-AG6P? Is it counter to humans, for instance? Why?

17) “Treatment of neutropenia and neutrophil dysfunction in GSD1b and G6PC3-deficient patients has not benefited from a classical clinical trial. This is ethically justified because the repurposing of empagliflozin to treat the neutropenia in both diseases has clearly improved neutrophil counts, neutrophil dysfunction, and the severity and number of infectious episodes, while this was often not the case under GCSF treatment.” I don’t really understand this sentence. Why is it ethical that patients have not benefited from a clinical trial if it is efficacious?

18) Line 548: This results not result.

19) Figure 4: The following text in the legend is not reflected in the figure itself: “In this case, the digestion of starch may occasionally produce a molecule of 1,5-anhydrofructose (rather than of glucose) if instead of using a water molecule in the breakdown of starch, alpha-glucosidases catalyze an elimination reaction. The 1,5-anhydrofructose molecule is quickly reduced to 1,5-AG by a ubiquitous NADPH-dependent 1,5-anhydrofruct"

The quality of the English language is satidfactory.

Author Response

The answers to the comment of reviewer 2 are in attachement

Reviewer 3 Report

This review article focuses on the relationship between neutropenia and two conditions, glycogen storage disease (GSD) 1b and glucose-6-phosphatase translocase (G6PC3) deficiency. The review provides an overview of the pathophysiological mechanisms underlying neutropenia in these conditions, which are related to impaired glucose metabolism and the accumulation of 1,5-AG6P, a potent inhibitor of hexokinases that intoxicates neutrophils by blocking glucose metabolism. The authors also discuss potential treatment options for neutropenia in these conditions, including SGLT2 inhibitors. Overall, the review provides valuable insights into the mechanisms underlying neutropenia in GSD 1b and G6PC3 deficiency and highlights potential treatment options for these conditions.

The study revealed that patients with glycogen storage impairment exhibit increased plasma and abnormal urine organic acid, leading to mitochondrial impairment and insulin resistance. The overload of mitochondria might produce by-products that could impact the signaling pathway. As glycogen storage impairment results in neutropenia, the authors should explore the link between insulin resistance and neutropenia.

In addition, the authors discussed the potential of SGLT2 inhibitors as a treatment for neutropenia. However, a paper published in 2015 by Yanay et al. (doi: 10.2147/JIR.S84993) showed that GLP1 agonists such as exendin-4 can decrease inflammatory cytokines and increase neutrophil count in a rat model of endotoxemia. The authors should consider discussing this finding in their review.

Author Response

The answers to the comments of reviewer 3 are in attachement
